# Two-dimensional antimonene single crystals grown by van der Waals epitaxy

Jianping Ji[1,*], Xiufeng Song[1,2,*], Jizi Liu[1,2,*], Zhong Yan[1,2,*], Chengxue Huo[1,*], Shengli Zhang[1,2,*], Meng Su[3], Lei Liao[3], Wenhui Wang[4], Zhenhua Ni[4], Yufeng Hao[5,6] & Haibo Zeng[1,2]

Unlike the unstable black phosphorous, another two-dimensional group-VA material, antimonene, was recently predicted to exhibit good stability and remarkable physical properties. However, the synthesis of high-quality monolayer or few-layer antimonenes, sparsely reported, has greatly hindered the development of this new field. Here, we report the van der Waals epitaxy growth of few-layer antimonene monocrystalline polygons, their atomical microstructure and stability in ambient condition. The high-quality, few-layer antimonene monocrystalline polygons can be synthesized on various substrates, including flexible ones, via van der Waals epitaxy growth. Raman spectroscopy and transmission electron microscopy reveal that the obtained antimonene polygons have buckled rhombohedral atomic structure, consistent with the theoretically predicted most stable β-phase allotrope. The very high stability of antimonenes was observed after aging in air for 30 days. First-principle and molecular dynamics simulation results confirmed that compared with phosphorene, antimonene is less likely to be oxidized and possesses higher thermodynamic stability in oxygen atmosphere at room temperature. Moreover, antimonene polygons show high electrical conductivity up to $10^4\,\mathrm{S\,m^{-1}}$ and good optical transparency in the visible light range, promising in transparent conductive electrode applications.

[1] Institute of Optoelectronics & Nanomaterials, Key Laboratory of Advanced Display Materials and Devices (Ministry of Industry and Information Technology), College of Materials Science and Engineering, Nanjing University of Science and Technology, Nanjing 210094, China. [2] Herbert Gleiter Institute of Nanoscience, Nanjing University of Science and Technology, Nanjing 210094, China. [3] Department of Physics and Key Laboratory of Artificial Mircro- and Nano-structures of Ministry of Education, Wuhan University, Wuhan 430072, China. [4] Department of Physics, Southeast University, Nanjing 211189, China. [5] Center for Integrated Science and Engineering & Department of Mechanical Engineering, Columbia University, New York, New York 10027, USA. [6] National Laboratory of Solid-State Microstructures and Department of Materials Science and Engineering, Nanjing University, Nanjing 210093, China. * These authors equally contribute to this work. Correspondence and requests for materials should be addressed to H.Z. (email: zeng.haibo@njust.edu.cn).

Many two-dimensional (2D) materials have been studied since the discovery of graphene in 2004 (ref. 1), yet elemental 2D materials are rather scarce. Other group-IVA (group-14, the carbon group) elemental 2D materials, such as silicene and germanene, have been successfully synthesized[2,3]. Black phosphorene (BP), as a typical group-VA (group-15, the nitrogen group) elemental 2D material, attracts intensive interests since 2014 (ref. 4). Very recently, several groups report the experimental synthesis of borophene[5,6], a group-IIIA (group-13, the boron group) elemental 2D material.

Graphene and the group-IVA analogues, silicene and germanene are zero-bandgap semimetals. Borophene is also reported to have metallic characteristics[5]. BP is the only semiconducting elemental 2D material reported so far, very attractive for novel applications in nanoelectronics and nanophotonics. However, the practical applications of BP are very challenging due to the severe unstability of BP in air[7–9]. Although bulk black phosphorus is the most stable allotrope of phosphorus, thin BP samples of 10-nm thickness may degrade in days whereas monolayer or few-layer BP may degrade within hours. Another issue that impedes the real applications of BP is the difficulty in sample synthesis. BP nanosheets can be exfoliated from bulk black phosphorus via Scotch tape method or liquid exfoliation method, while bulk black phosphorus is obtained from red phosphorus under high pressure (10 kbar), high temperature (1,000 °C) conditions. The direct synthesis of atomically thin BP is still a great challenge.

In contrast to BP, another group-VA element composed 2D material, antimonene, was recently predicted to be of good stability, as well as extraordinary properties through first-principle calculations[10–13]. Zhang et al.[10] further predicted that among several possible allotropes of monolayer antimonene, the rhombohedral phase, ususally viewed as ABC stacking of monolayer antimonene (β-phase), has the best stability. They also predicted that the band structures of antimony would transfer from semimetal in the bulk form into semiconductor when thinned to one atomic layer. According to theoretical calculations, anitmonene presents high carrier mobility[11], superior thermal conductivity[14], strain induced band transition[10,15] and promising spintronic properties[16]. Despite the rapid progress of theoretical works about anitmonene mentioned above, reports of experimental studies on antimonene is still scarce[17–19]. Early this year, Lei et al. realized the synthesis of monolayer antimoene on Sb₂Te₃ and Bi₂Te₃ by molecular beam epitaxy[17]. However, the products are small size domains, which hampered the further research. Antimonene flakes was obtained via mechanical exfoliation, but the quantity is very little[18]. Moreover, Tsai et al. claimed the advent of noncontinuous antimonene film composed of multilayer antimonene with about 20 nm thickness and observed the orange light emission, which was attributed to quantum confinement effect and turbostratic stacking order[19]. The experimental investigations on the extraordinary properties of antimonene and the various applications significantly rely on the reproducible synthesis of high-quality, single crystalline samples. Due to the tremendous difference between various possible allotropes of antimonene, the identification of the exact atomic structure of synthesized anitmonene samples is also very important. In this case, the experimental synthesis of high-quality and monocrystalline antimonene together with systematic atomic structure characterizations is very urgent for the emerging research hotspots of group-VA elemental 2D materials.

Here, we demonstrate that high-quality, few-layer antimonene monocrystalline polygons can be synthesized on various substrates, including flexible ones, via van der Waals epitaxy growth, and highlight their stable β-phase as previously predicted. The few-layer antimonene polygons are single crystals of large

lateral size (5–10 μm) and thin thickness (1–50 nm). Raman spectroscopy and transmission electron microscopy (TEM) reveal that the obtained antimonene polygons have buckled hexagonal atomic structure, consistent with the theoretically predicted most stable allotrope (β-phase). The obtained anitmonene layers exhibit good stability when exposed in air, which can be verified by optical microscopy, atomic force microscopy (AFM), Raman spectroscopy, energy dispersive spectrometer (EDS) and X-ray photoelectron spectroscopy (XPS). First-principle calculations reveal that compared with phosphorene, antimonene is less likely to be oxidized in air. First-principles calculations and molecular dynamics simulations confirm much better thermodynamic stability of antimonene than phosphorene in oxygen atmosphere at room temperature. We also find that antimonene polygons possess high electrical conductivity up to $10^4$ S m$^{-1}$ and good optical transparency in the visible light range, which in-principle can be applied in flexible transparent conductive electrode applications. This work provides experimental evidence of the successful synthesis of few-layer antimonene monocrystalline polygons and it may pave the way for the further experimental investigations of the unique properties of antimonene.

## Results

**Synthesis of antimonene single-crystal polygons.** In conventional epitaxy growth techniques, the production of high-quality epitaxial layer requires that the substrate and the epitaxial layer must have the same symmetry and very close lattice constants. Van der Waals epitaxy first introduced by Koma et al.[20] can almost avoid these restrictions and it has been proven as a facile technique in synthesizing 2D materials and their vertical heterostructures[21–25]. Van der Waals epitaxy utilizes materials without dangling bonds on their surface as substrate and the epitaxial layers are connected with the substrates through weak van der Waals forces instead of strong chemical bonding. Therefore van der Waals epitaxy enables the epitaxial growth of layered materials with different crystalline symmetry to the substrate. In addition, van der Waals epitaxy can be achieved even when the lattice constant mismatch is as high as 50% and no excessive strain exists in the epitaxial layer. Van der Waals epitaxy has been successfully applied to grow various layered materials including topological insulators Bi₂X₃ (X = Se or Te) (ref. 26), transition-metal dichalcogenide[21], 2D GaSe[22] and tellurium polygons[27].

In this work, few-layer antimonene polygons were synthesized on fluorophlogopite mica (KMg₃(AlSi₃O₁₀)F₂) substrates with exposed (001) surface via van der Waals epitaxy. The sample synthesis procedure is shown in Fig. 1a. A two-zone tube furnace with separate temperature controls was used. Commercial antimony powder placed in the source zone ($T_1$) was heated up to 660 °C to provide antimony vapor. The fluorophlogopite mica substrates were placed in the downstream area with temperature $T_2 = 380$ °C. The synthesis process was maintained for 60 min. Then the furnace was cooled down to room temperature. More details of the sample synthesis process are provided in the Methods section.

Figure 1b illustrates the van der Waals epitaxy of antimonene layers on mica. Mica substrate was found to be very suitable for van der Waals epitaxy growth owning to the absence of dangling bonds on the ultra-smooth surface[20–22,27]. The migration energy barrier of antimony atoms on mica substrate is very small, which results in a high migration rate along the mica substrate and a fast lateral growth of 2D antimonene polygons. The van der Waals epitaxy growth feature was further confirmed by growing antimonene polygons on various substrates under same conditions as that on mica. Silicon and sapphire

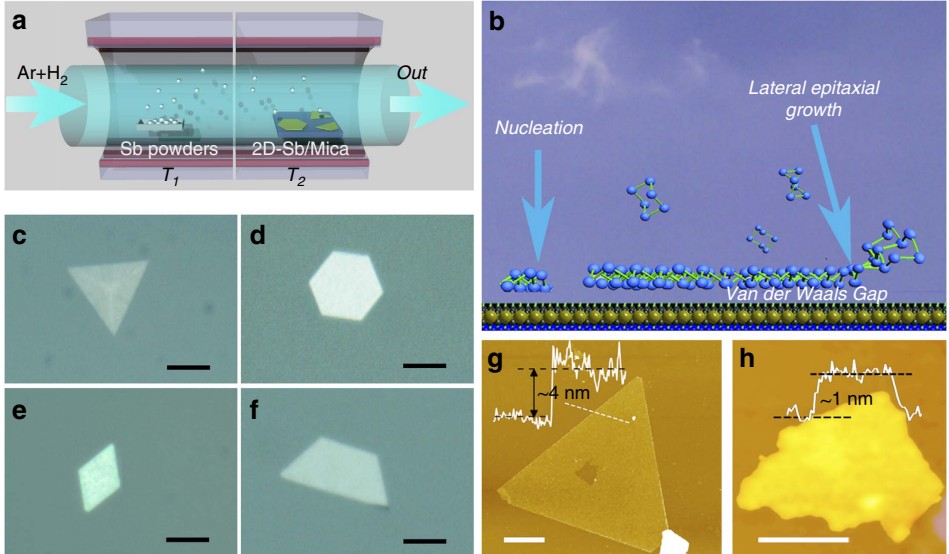

**Figure 1 | Antimonene polygons synthesized on mica substrates via van der Waals epitaxy.** (**a**) Schematic illustration of the sample synthesis configurations. (**b**) Schematic diagram of van der Waals epitaxy. (**c–f**) Optical images of typical antimonene polygons with triangular, hexagonal, rhombic and trapezoidal shapes, respectively. The scale bar is 5 μm. (**g**) AFM image of a typical triangular antimonene sheet. The thicknesses are 4 nm. The scale bar is 1 μm. (**h**) AFM image of a tiny antimonene sheet. The thicknesss is ca. 1 nm. The scale bar is 50 nm.

substrates have been tested. A large number of irregular microstructures can be observed on optical microscope images in Supplementary Fig. 1. Thus, the absence of dangling bonds on the substrate surface is critical for the successful growth of antimonene layers, which was further confirmed via XPS. As shown in Supplementary Fig. 2, peaks at 528.4 and 538 eV are attributed to Sb $3d_{5/2}$ and $3d_{3/2}$, respectively, characteristic of nonvalent antimony[19]. This indicates nonexistence of chemical bonding between antimonene layers and mica, which is consistent with the universal characteristics of van der Waals epitaxy[20].

Figure 1c–f shows typical optical microscope images of the few-layer antimonene sheets synthesized on the substrate with scale bar of 5 μm. It is interesting to notice that those antimonene sheets exhibit several types of polygonal shapes, including triangles, hexagons, rhombus and trapezoids. The well-defined shapes of those polygons are the first indication of good crystallinity. Besides, we can also find mutilayer antimonene ribbons, as shown in Supplementary Fig. 3 and part of them are confirmed to be nanoribbons. The optical microscope image of ribbons in Supplementary Fig. 3 reveals that the obtained antimonene ribbons are actually hexagons extremely stretched along one direction. Most polygons have lateral sizes around 5–10 μm. The AFM was carried out to measure the thickness of antimonene polygons. As indicated in Fig. 1g, antimonene polygons have thicknesses down to 4 nm (around 10 number of atomic layers). In fact, we do also find very tiny sheet with lateral size around 100 nm and thickness down to 1 nm (Fig. 1h), which is characteristic of monolayer antimonene. As illustrated in Supplementary Fig. 4, one tiny sample with a similar scale was proved to be antimony via EDS.

To further investigate the growth process of the layered antimonene, we studied the growth process under different durations. Typically, as shown in Fig. 1b, crystal growth on mica is divided into nucleation and lateral growth. At the initial stage, hot antimony vapours,carried by Ar/H2 gas, cool down and deposit on the mica substrate, resulting in the formation of nuclei. Subsequently, due to the low migration barrier energy, adatoms on mica migrate fast to the edge of initial nuclei, which grow fast along along the chemically passivated surface into layers. Both the nucleation and lateral growth are experimentally confirmed via

atomic force microscopy, as shown in Supplementary Fig. 5. Moreover, during the growth period, the average lateral size of samples under different durations were measured and counted. As the Supplementary Fig. 5 shown, the growth have finished in the first 10 min, even for longer durations. And, the growth rate of antimonene layers during the first 10 min is calculated to be 0.5 μm min$^{-1}$.

**Atomic scale microstructure of antimonene single-crystal polygons.** Bulk antimony has several allotropes, among which the most stable form has the rhombohedral structure (β-phase), similar to the structure of grey arsenic. The atomic structure of β-phase antimony is illustrated in Fig. 2a. Monolayer β-phase antimonene consists of buckled hexagonal rings composed of Sb atoms connected through sp$^3$ bonding. Moreover, bulk antimony can be viewed as the ABC stacking of monolayer antimonene[10,11,13,14]. Another common allotrope of antimony has the orthorhombic structure (α-phase), similar to that of black phosphorus displayed in Supplementary Fig. 6. Bulk α-phase antimony can be viewed as AB stacking structures of α-phase antimonene. Theoretical calculations predicted that both α-phase and β-phase antimonene are stable[13,14]. Other possible allotropes of monolayer antimonene have also been studied theoretically, among which the β-phase antimonene has the lowest average binding energies[11]. Therefore, in principle, the synthesized antimonene will prefer the β-phase structure.

To identify the chemical composition and crystal structure of our synthesized antimonene layers, high-resolution transmission electron microscopy (HRTEM), scanning transmission electron microscopy (STEM), energy dispersive spectroscopy (EDS) and selected area electron diffraction (SAED) are employed. Fig. 2b shows a low-magnification high-angle annular dark-field (HAADF)-STEM image of a polygonal antimonene layer. EDS mapping images in Fig. 2c,d indicate that the crystal is composed of Sb and a small amount of O. According to the corresponding EDS analysis in Fig. 4b, there is ca. 6 wt% oxygen, which is attributed to contamination during the process of transferring, since the value is much lower than the oxygen content in antimony oxide.

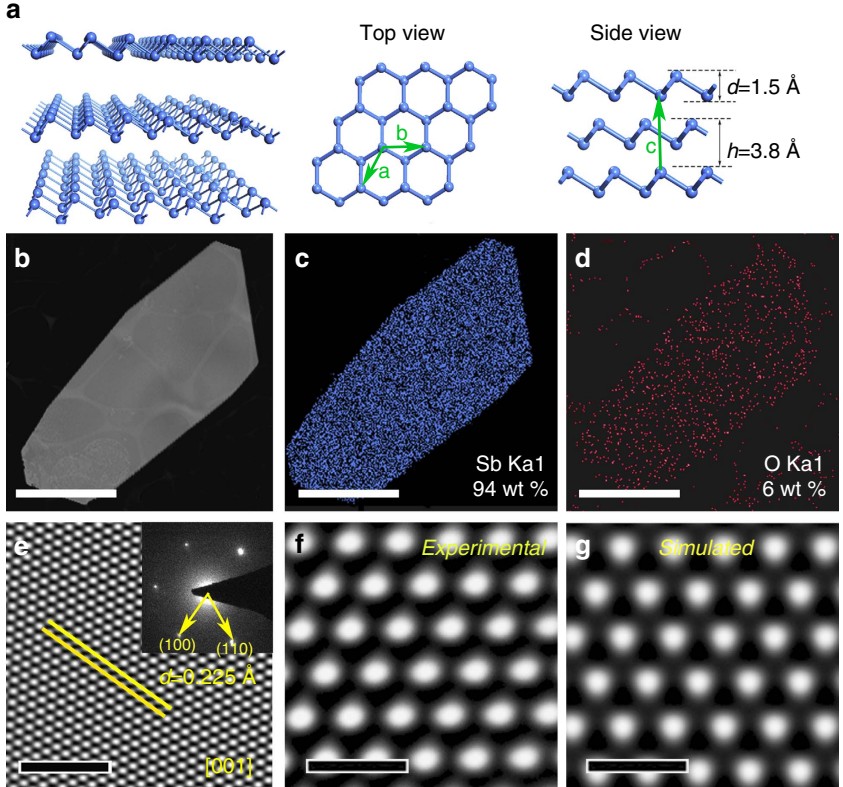

**Figure 2 | Atomic structure of synthesized antimonene polygons.** (**a**) Schematic illustrations of the atomic structure of β-phase antimonene. (**b**) Low-magnification TEM image of a well shaped antimonene sheet transferred onto copper grid. The scale bar is 1 um. (**c**) Sb element mapping. Sb content is measured to be 94 wt.%. The scale bar is 1 um. (**d**) O element mapping. O is measured to be 4%. The scale bar is 1 um. (**e**) HRTEM image. Inset: SAED pattern along the [001] zone axis. The scale bar of the HRTEM image is 2 nm. (**f**) Enlarged image of few-layer antimonene in **e**. The scale bar is 0.5 nm. (**g**) Simulated HRTEM image of few-layer antimonene. The scale bar is 0.5 nm.

To explore the structure of the 2D crystal, we provide the simulated kinematical SAED pattern of α-phase (along [001] zone axis) and β-phase (along [001] zone axis) in Supplementary Fig. 7. It clearly demonstrates β-phase can be distinguished from α-phase by comparing SAED patterns. The experimental SEAD pattern shown in Fig. 2e strictly confirms the structure of β-phase viewing along [001] zone axis, indicating that the 2D Sb crystal belongs to rhombohedral structure (β-phase) rather than orthorhombic structure (α-phase). Comparing the experimental and simulated SAED diffraction spot, the derived d spacing of (100) was measured to be 0.225 nm, consistent with the previous results[18].

The corresponding typical HRTEM image is also provided in Fig. 2e, extracted from which, an enlarged image is displayed in Fig. 2f. The HRTEM image with atomic resolution was taken under the condition of $Cs = -13\,\mu m$ and $\Delta Z = +7\,nm$. Under this condition, atom columns appear bright points on a dark background. For comparison, with the same microscope parameters, the simulated image of few-layer antimonene with ABC stacking is shown in Supplementary Fig. 8. Obviously, there is very good agreement on the arrangement of bright points between the experimental and the simulated atomic images, which confirms the ABC stacking of the synthesized antimonene layers. As shown in Supplementary Figure 9, the calculated atomic images of antimonene with different stacking are apparently different with each other. Above results verify that the synthesized antimonene polygons adopt rhombohedral structure (β-phase) as predicted by theoretical calculations.

**Phase structure of monocrystalline antimonene layers**. To further study the crystal structure and quality of synthesized antimonene polygons, Raman spectroscopy was carried out. Fig. 3a compares the Raman spectra of typical few-layer antimonene with bulk antimony (β-phase) and antimony trioxide. Two Raman peaks, $E_g$ at $\sim 111\,cm^{-1}$ and $A_{1g}$ at $\sim 149\,cm^{-1}$ can be seen in the Raman spectrum of few-layer antimonene. The $E_g$ modes are doubly degenerate in-plane vibrational modes and $A_{1g}$ is an out-of-plane vibrational mode. This Raman spectrum is characteristic of β-phase antimony, which agrees well with previous observations in β-phase bulk antimony[28]. And the Raman spectrum of few-layer anitmonene is completely different from that of antimony trioxide, which indicates that the synthesized atomically thin antimonene does not oxidize when exposed in air during the sample preparation and characterization process.

Since α-phase antimonene has not been synthesized yet, no experimental Raman data can be provided for direct comparison. However, the atomic structure of α-phase antimonene is similar to BP. The primitive cell of α-phase antimonene contains four atoms leading to twelve vibrational modes, among which there are six Raman active modes. Using density functional theory (DFT) calculations, we simulated the Raman peak frequencies of α-phase monolayer antimonene in Supplementary Fig. 10 (Supplementary Information ). The details of the simulation method are provided in Methods section.

Compared with the Raman spectrum of bulk antimony, both $E_g$ and $A_{1g}$ peak frequencies of β-phase few-layer antimonene move to the higher wavenumber region (also called blueshift).

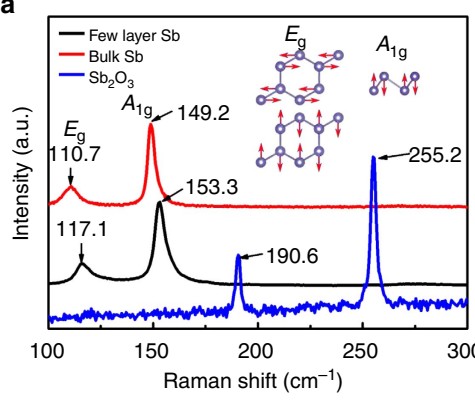

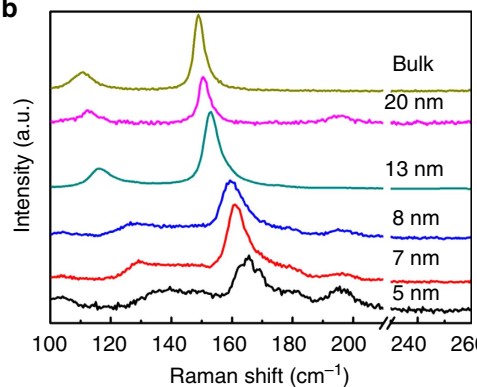

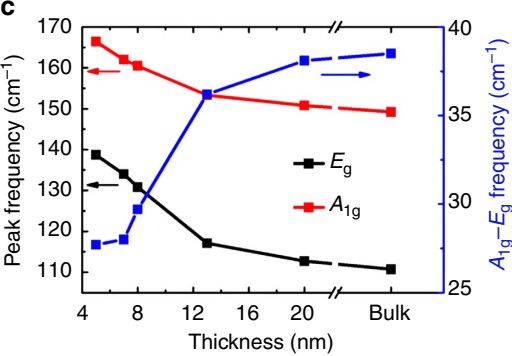

**Figure 3 | Raman spectroscopy of antimonene polygons.** (**a**) Raman spectra of bulk antimony (β-phase), few-layer antimonene and antimony trioxide. Inset: the vibrational modes of β-phase antimonene. (**b**) Raman spectra of antimonene polygons with thicknesses varying from 5 nm to bulk. (**c**) $A_{1g}$, $E_g$ peak frequencies (left vertical axis) and the energy difference (right vertical axis) of those two peaks plotted against sample thickness. The energy difference between the $A_{1g}$ and $E_g$ peaks increases monotonically with sample thickness.

Figure 3b shows Raman spectra of antimonene polygons on mica substrate with thickness varying from 5 nm to bulk. The thicknesses of the tested polygons were measured by AFM separately. Both $E_g$ and $A_{1g}$ peaks can be observed in the Raman spectra of antimonene polygons with different thicknesses. And there is a clear trend that both $E_g$ and $A_{1g}$ peak frequencies move to the higher wavenumber region, when the thickness of sheets decreases. When the sheet thickness decreases, Raman signals of the mica substrate emerge, illustrated in Supplementary Fig. 11 (Supplementary Information). We can see two additional Raman peaks at ∼100 and ∼190 cm$^{-1}$ in the Raman spectrum of 5-nm thick antimonene polygon. The 190 cm$^{-1}$ Raman peak

does not come from the oxidation of antimonene polygons, otherwise another Raman peak will emerge at 255 cm$^{-1}$ with higher intensity.

The Raman peak frequencies of $E_g$ and $A_{1g}$ modes are fitted by Lorentz function and the fitting results are plotted in Fig. 3c. When the sample thickness decreases from bulk to 5 nm, the $E_g$ peak frequency increases from 110.7 to 138.7 cm$^{-1}$ and the $A_{1g}$ peak frequency increases from 149.2 to 166.4 cm$^{-1}$. Theoretical calculations predict that the $E_g$ and $A_{1g}$ peak frequencies of monolayer β-phase antimonene are at 150 and 195 cm$^{-1}$, respectively[14], which are consistent with the blueshift trend found in our studies when the sheet thickness decreases. Similar blueshift trends of Raman peak frequencies with the thickness decreases have been reported in the Raman studies of other 2D materials[29,30]. These shifts are probably attributed to the lattice constant shrinks when the number of layers decreases or long-range Coulombic interlayer interactions[29,30]. The frequency differences between $A_{1g}$ peak and $E_g$ peak are plotted against sample thicknesses in Fig. 3c. There is a monotonic decrease trend of this frequency difference when the sample thickness decreases. The frequency difference of bulk antimony is 38.5 cm$^{-1}$ and decreases to 27.7 cm$^{-1}$ when thickness decreases to 5 nm. This relation can be utilized as a facile tool to estimate the thickness of antimonene polygons.

**High stability of mutilayer antimonene.** As is known to all, the stability in air significantly affects further study and application prospects. It's a typical negative example that the practical applications of BP are very challenging due to the severe unstability of BP in air[7–9]. Therefore, it's urgent to verify the stability of antimonene in air.

Figure 4a illustrates the optical microscopy, AFM and Raman spectroscopy measurements of one freshly-made mutilayer antimonene and those obtained after one month aging. Optical images indicate the stability since the specific flake remains unchanged, as shown in Fig. 4a. Both $A_{1g}$ and $E_g$ peak in the Raman spectra of one freshly-made antimonene triangle have a good consistence with that stored for one month. Moreover, the characteristic Raman spectrum of antimonene possesses no additional peaks from antimony trioxide or antimony pentaoxide, which further confirms the good stability of few-layer antimonene. Surface roughness, measured via AFM, is another common index to evaluate sample stability. One-month exposure to the air results in no obvious change in the low surface roughness, indicating much higher stability than black phosphorus. Moreover, EDS spectra in Fig. 4b further verify the stability while the O contamination is attributed to the hash transfer. As shown in Supplementary Fig. 2, X-ray photoelectron spectra also indicate the stability of antimonene. Specifically, the Sb 3d spectrum measured after the sample exposed in air for one month are well fitted with a spin-orbit doublet, characteristic of nonvalent antimony, highly consistent with the freshly-made one.

The observed outstanding stability of 2D antimonene in air revealed by above results will greatly advance the future experimental exploration on basic properties and potential applications. But, what's the reason behind their ultrahigh stability, especially when compared with another 2D material in the same group, phosphorene?

The thermodynamic and kinetic simulations on stability were presented in Fig. 4c,d based on first principles calculations and molecular dynamics calculations. On the one hand, the interactions of $O_2$ with antimonene and phosphorene were compared according to their reaction energy. First, we relax antimonene and phosphorene configures in the system containing oxygen molecules, and then the most stable antimonene oxide ($O=Sb_{2D}=O$) and phosphorene oxide

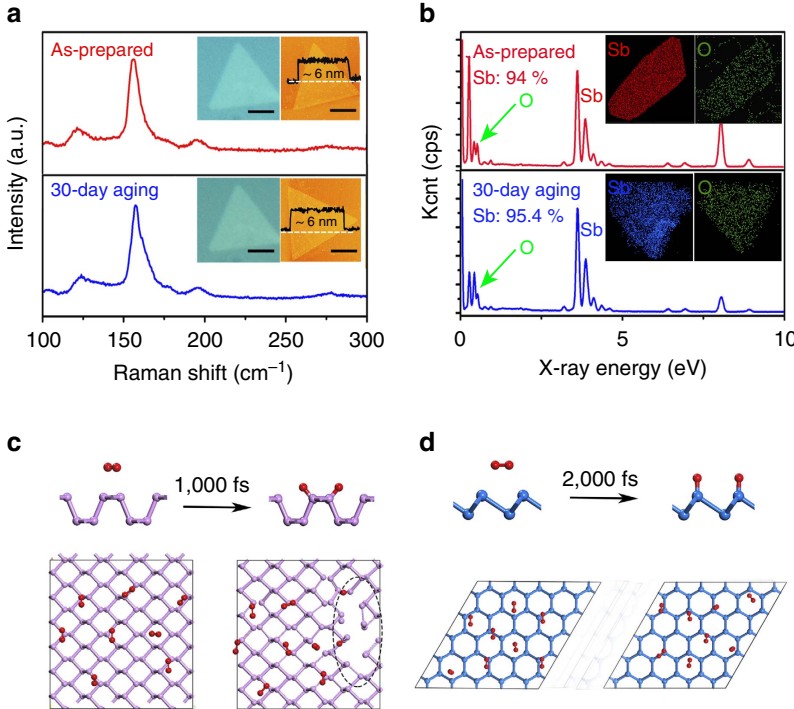

**Figure 4 | Stability verification via experiments and simulations of antimonene compared with BP.** (**a**) Optical images, AFM images and Raman spectra of antimonene layers before and after a 30-day aging. The scale bar is 2 um. (**b**) EDS analysis of antimonene sheets randomly selected om the copper grid before and after a 30-day aging. (**c**) Atomic structure of phosphorene oxide (O = P$_{2D}$ = O) and time-dependent snapshots of the configurations revealing interaction of O$_2$ with phosphorene. (**d**) Atomic structure of phosphorene oxide (O = Sb$_{2D}$ = O) and time-dependent snapshots of the configurations revealing interaction of O$_2$ with antimonene.

(O = P$_{2D}$ = O) structure can be obtained, as shown in Fig. 4a,b, respectively. So, the oxidizing reaction as shown in Equation (1) and (2) emerge, and the related energy can be extracted. Significantly, the exothermic energies Q corresponding to the O$_2$ dissociation on phosphorene and antimonene are $-4.15$ and $-1.01$ eV/O$_2$, respectively. Note that the triplet state of O$_2$ is used to calculate the dissociation energy, and the method has been well confirmed in phosphorene oxidation[31]. Such smaller exothermic energies strongly support that O$_2$ are more likely to react with phosphorene than antimonene, evidencing higher air stability of antimonene.

$$Sb_{2D} + O_2 \rightarrow O = Sb_{2D} = O, \quad Q = -1.01 \, \text{eV/O}_2 \quad (1)$$

$$P_{2D} + O_2 \rightarrow O = P_{2D} = O, \quad Q = -4.15 \, \text{eV/O}_2 \quad (2)$$

On the other hand, first-principles molecular dynamics simulations were used to further check the thermodynamic stability of antimonene. To approach the real stability, we adopt a very large supercell by $5 \times 5$ and $6 \times 4$ repeated units for antimonene and phosphorene, respectively. Figure 4c,d shows their time-dependent snapshots, revealing interaction of O$_2$ with antimoenen and phosphorene at 300 K. Initially, we placed a few molecules with about 4 Å distance above the surfaces of the relaxed antimonene and phosphorene. For the case of phosphorene, some O$_2$ molecules will first move closer to the native phosphorus atoms, then easily dissociate into O atoms. Only after 1,000 fs, phosphorene is seriously disrupted, as shown in Fig. 4c. However, for the case of antimonene, O$_2$ still did not dissociate into oxygen atoms after 2,000 fs, and the sheet is well kept in its original configuration. The above two aspects fully proved that 2D antimonene has very good thermodynamic stability and can maintain its structural integrity during O$_2$

atmosphere. This conclusion further confirmed the above experiment results.

**Flexible transparent conductivity of antimonene polygons.** We next fabricated thin film transistors on antimonene polygons with hafnium oxide (HfO$_2$) as top gate dielectric. The devices were fabricated directly on mica substrates after sample synthesis. The schematic of the device structure is shown in Fig. 5a. The electrical contacts were defined by standard electron beam lithography followed by metal deposition of 10-nm-thick chromium and 30-nm-thick gold. A layer of 15-nm-thick HfO$_2$ as the dielectric for the top gate was then grown by atomic layer deposition. Finally, one layer of 50-nm-thick gold was deposited as the top gate electrode. Figure 5b shows the optical image of a typical antimonene device. The antimonene sheet is in triangle shape with thickness around 30 nm.

The electrical characterization of fabricated antimonene devices were performed using a semiconductor parameter analyser at room temperature. Figure 5c shows the current–voltage ($I_{ds}$–$V_{ds}$) characteristics of three typical devices with antimonene thicknesses of 30, 40 and 50 nm, respectively, measured at zero gate bias. The I–V curves show good ohmic behaviour. The electrical resistance of the 30-nm thick antimonene device is calculated to be 600 Ω. The device channel length, width and thickness are 1 μm, 3.5 μm and 30 nm, respectively. Thus, the electrical conductivity of the synthesized antimonene sheet is calculated to be $1.6 \times 10^4$ S m$^{-1}$, a typical conductivity value for semimetals. This result is also consistent with our theoretical predictions, since the semimetal to semiconductor transition will occur only when the thickness of antimonene is thinned to one atomic layer. The inset in Fig. 5c shows the $I_{ds}$–$V_g$ curve of the 30-nm thick antimonene device

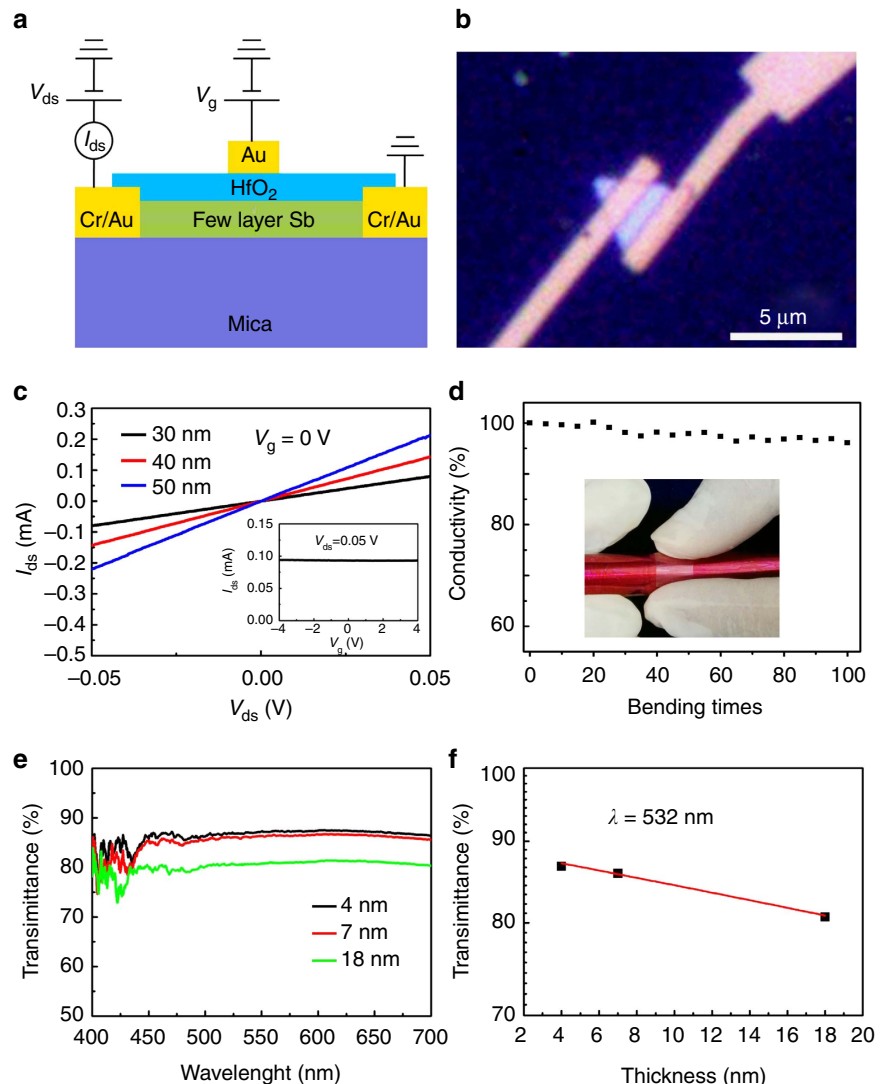

**Figure 5 | The flexible transparent conductivity of antimonene polygons.** (**a**) Schematic illustration of top-gated antimonene thin film transistors. (**b**) Optical image of a typical device fabricated on a triangular antimonene sheet. (**c**) $I_{ds}$–$V_{ds}$ characteristics of three tested devices acquired at $V_g = 0$ V. Inset: $I_{ds}$–$V_g$ curve of the 30-nm thick antimonene device acquired at 0.05 V bias voltage. (**d**) Electrical conductivity of the tested device after bending. Inset: Photograph of bended antimonene devices on mica substrate. (**e**) Transmittance spectra of three typical antimonene polygons of 4, 7 and 18 nm thicknesses, respectively. (**f**) Transmittance of antimonene polygons versus sample thickness.

measured at fixed source-drain bias, $V_{ds} = 0.05$ V. Due to the metallic properties of antimonene channel, no gating effects could be observed when the gate voltage swept from $-4$ to 4 V. The antimonene devices fabricated on flexible mica substrate also possess high flexibility. The antimonene/mica sample was attached onto the smooth surface of an iron rod to ensure the bending effect was applied. We measured the electrical conductivity variations of the 30-nm thick antimonene device as shown in Fig. 5b after continuously bending the device for 100 times and the results are shown in Fig. 5d. Data points in this curve were obtained by measuring the $I_{ds}$–$V_{ds}$ characteristics after bending the device for every five times until the total 100 times bending. Electrical conductivity only deceased by 4% after 100 times bending compared with the initial data before bending.

We also find that the synthesized antimonene polygons with nanometre thickness are of good transparency in the visible light range. The transmittance spectra of three typical antimonene polygons in the wavelength range from 400 to 700 nm are shown

in Fig. 5a. The transmittance was calculated using the transmitted light intensity through antimonene polygons on mica substrate divided by the one measured through blank mica substrate (see more details in the Methods section). We find that the transmittance of antimonene polygons is practically independent of wavelength in the range from 450 to 700 nm. The transmittance at 532 nm verses sample thickness is plotted in Fig. 5b. The transmittance remains over 80% for antimonene polygons of 18 nm thickness. The good electrical conductivity of antimonene combined with the good optical transparency in the visible light range might lead to interesting applications in the transparent conductive electrode related areas.

## Discussion

In summary, we synthesized high-quality, few-layer antimonene polygons on mica substrate through van der Waals epitaxy. HRTEM microscopy and Raman spectroscopy revealed that the obtained antimonene polygons exhibit buckled hexagonal structure

(β-phase), which is consistent with the atomic structure of the most stable allotrope of monolayer antimonene predicted by the previous theoretical studies. Optical microscopy, AFM, Raman spectroscopy and XPS also proved the good stability of antimonene polygons when exposed in air. First-principle calculations reveal that compared with phosphorene, antimoene is less likely to be oxidized in air. And first principles molecular dynamics simulations confirm much better thermodynamic stability of antimonene than phosphorene in oxygen atmosphere at room temperature. Electrical characterizations demonstrate that the synthesized antimonene polygons have good electrical conductivity on the order of $10^4\,S\,m^{-1}$. Combined with the wavelength independent high transparency in the visible light range, antimonene is expected to have potential applications as flexible transparent conductive electrodes. This work paves the way for further experimental investigations on the extraordinary properties of antimonene, as well as various applications.

## Methods

**Synthesis of antimonene polygons.** The antimonene polygons were grown via van der Waals epitaxy method using a two zone tube furnace with separate temperature controls. 100 mg of commercial antimony powder was placed in the source zone and the fluorophlogopite mica substrates were placed in the downstream area. During the synthesis process, the antimony powder was heated up to $T_1 = 660\,°C$ and vaporized. The vapor of Sb atoms was carried downstream by mixed $Ar/H_2$ gas (30% $H_2$) with constant flow rate of 700 sccm. The temperature of the area containing mica substrates was fixed at $T_2 = 380\,°C$ and the Sb atoms were deposited on mica substrates and grown into various polygons. The synthesis process was maintained for 60 min. Then the furnace was naturally cooled down to room temperature.

**Transfer of antimonene polygons.** A toluene solution (9 wt%) of polystyrene (280 K, Aldrich) was spin-coated on the mica substrate with antimonene polygons on top. After baked at 80 °C, the mica substrate was soaked in water droplet followed by being poked from the polystyrene (PS) layer edge. The PS layer attached to the antimonene polygons was scratched off rapidly. Subsequently, the PS-antimonene assembly was picked up with a tweezer and transferred onto other substrate. After air drying at 60 °C, the sample was baked at 150 °C to eliminate wrinkles on the transferred antimonene polygons. Finally, the sample was soaked in hot toluene for several minutes to remove PS layer and antimonene polygons were left on the substrate.

**Characterization of antimonene polygons.** The optical pictures were obtained with an optical microscope (Olympus, BX51). The thickness and surface topology was characterized using atomic force microscope (AFM, Bruker Multimodel-8). The TEM experiments were carried out using Titan T20 and G2 (for the HRTEM measurement). Raman measurements were performed with a Horiba Labram HR800 Raman spectrometer with an excitation wavelength of 532 nm. The transmittance spectra were measured using a halogen lamp (400–1,050 nm) as the light source and a NOVA-EX spectrometer (325–1,100 nm) as the light intensity detector. The transmittance was calculated using the transmitted light intensity through antimonene polygons on mica substrate divided by the one measured through blank mica substrate.

**Stability simulations of antimonene.** First-principles calculations are performed by DFT within the generalized-gradient approximation implemented in the DMol software[32]. The Perdew, Burke and Ernzerhof exchange-correlation functional is used, and all structures are without any symmetry constraints with the Grimme dispersion correction[33]. The global cutoff radius is set to be 5.5 Å. The convergence criteria applied during geometry optimization is $1.0 \times 10^{-6}$ Hartree for energy. First-principles molecular dynamics simulations in the NVT ensemble are carried out with a time step of 1.0 fs. We adopt $5 \times 5$ and $6 \times 4$ supercell with a vacuum of about 25 Å. The temperature was controlled using the Nosé–Hoover method[34].

**Fabrication and characterization of antimonene devices.** The electrical contacts were defined by standard electron beam lithography followed by metal deposition of 10-nm-thick chromium and 30-nm-thick gold. A layer of 15-nm-thick $HfO_2$ as the dielectric for the top gate was then grown by atomic layer deposition. Finally, one layer of 50-nm-thick gold was deposited as the top gate electrode. I–V characteristics of fabricated antimonene devices were measured with Agilent 4,156 semiconductor parameter analyser at room temperature.

**Data availability.** The data that support the findings of this study are available from the authors on reasonable request, see author contributions for specific data sets.

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

## Acknowledgements

This work was financially supported by the National Basic Research Program of China (2014CB931702), NSFC (51572128), NSFC-RGC (5151101197), NSFC (51502140), NSFC (21403109), Natural Science Foundation of Jiangsu Province (BK20140769, BK20150761), the Fundamental Research Funds for the Central Universities (No. 30916015106), the Fundamental Research Funds for the Central Universities, and PAPD of Jiangsu Higher Education Institutions.

## Author contributions

H.Z. designed the idea and protocol of this work. J.J. implemented the experiments of growth, characterizations and performance measurements. X.S implemented Raman spectral measurements. J.L. implemented HRTEM characterization and analysis. Z.Y. and J.J. analysed atomic structure and wrote the draft. C.H., M.S., L.L., W.W. and Z.N. fabricated and analysed the FET devices. S.Z. implemented the simulation on the electronic and atomic structures. All authors discussed and commented on the manuscript.

## Additional information

**Competing financial interests:** The authors declare no competing financial interests.

