## [Peer Review File · Nature Communications]

Reviewers' comments:

Reviewer #1 (Remarks to the Author):

In this paper, the authors demonstrated that high-quality, few-layer antimonene single crystal polygons can be synthesized on various substrates via van der Waals epitaxy growth, and highlighted their strong stability in air. This is an interesting work containing some value and novelty. However, the following comments should be properly addressed before considering for publication in Nature Communications.

Q1: The authors claimed that antimonene nanoribbon can be found in Figure 1b. However, the width of the sample is even larger than 10 μm , how can it be called "nanoribbon"?

Q2: The authors mentioned that bulk antimony can be viewed as ABC stacking of monolayer antimonene. What is the stacking order of the few-layer antimonene in this manuscript? In addition, the crystal constants of few-layer antimonene should also be given.

Q3: The antimonene sheets in Figure 1g and Figure S2 are the same sample, why the thickness of the former is ca. 4 nm, while the later is 1 nm? Did you put the picture wrong in Figure S2?

Q4: The authors pointed out that the absence of dangling bonds on the substrate surface is critical for the successful growth of antimonene polygons. However, one antimonene sheet grown on mica in Figure S2 shows irregular shape. Is it oxidized or formed by some other reasons?

Q5: The authors mentioned that the migration energy barrier of antimony atoms on mica substrate is very small, resulting in a high migration and fast growth rate. Are there any references or calculation results to support this point? In addition, after 60-min growth, the lateral size of the sample is only ca. 10 μm . Such growth rate can hardly be called "fast".

Q6: The characterization of the stability of the antimonene sheet is not convincing only by Raman analyses. Optical, AFM and TEM EDX measurements of the freshly-made antimonene sheet and that stored one month should be given.

Q7: In the calculation part, the unit in the text is "ps", while it is "ns" in Figure 4. What is the real unit?

Q8: The V_{ds} in the inset of Figure 5c is 0.05V, while it is 0.5V in the text. Which one is the real measurement condition?

Reviewer #3 (Remarks to the Author):

This is interesting paper which reports synthesis and characterization of a new 2D material, antimonene. Stability of antimonene in air, as revealed by Raman spectra, suggests that antimonene could be an interesting 2D material for applications at nanoscale. The authors have used multi-layered antimonene to demonstrate its applicability for the flexible electronic device.

The following points should be addressed before accepting the paper for publication:

i) Page 7. The authors state that "The migration energy barrier of antimony atoms on mica substrate is very small, which results in a high migration rate along the mica substrate and a fast lateral growth of 2D antimonene polygons."

Is there any supporting references or data for this statement?

ii) Figure 4. Did the authors run ab-initio calculations up to 2000 ps? In the main text, the authors state that the simulation is up to 2000 ps, while in Figure 4, it is 2000 fs.

iii) Reference 25 predicts α and β - phases to be degenerate. Can the authors comment on it?

iv) The results of a recent paper for oxygen in phosphorene should be bench-marked with the results obtained previously (Wang et al. 2D Mater. 3, 025011, 2016).

In the present study, the calculated dissociation energy of O₂ on phosphorene is high. Did the authors use triplet or singlet O₂ to get the dissociation energy?

v) The dissociation energy of O₂ on antimonene is calculated to be -1.62 eV/molecule, which implies that O₂ prefers to dissociate on antimonene.

XPS measurement could confirm whether O₂ is dissociated or not. Can the authors address this discrepancy?

iii) Pl correct the citation for references and figures in the main text.

(a) Reference 13 -> Reference 25.

Reference 25 -> Reference 13, 25.

(b) Page 10. "Raman spectra of typical antimonene polygons stored in air for one month after sample synthesis are shown in Figure S5."

Figure S5 is not Raman spectra..

(c) Page 8. "The selected area electron diffraction (SAED) pattern shown in the inset of Figure 3c further confirms the hexagonal structure and its single crystal feature..."

Figure 3c -> Figure 2c.

(iv) Text: DFT calculations are not 'ab-initio' calculations.

ab initio -> first principles

Point-by-point replies to the reviewers' comments

Reviewer #1

Q1: The authors claimed that antimonene nanoribbon can be found in Figure 1b. However, the width of the sample is even larger than 10 um, how can it be called "nanoribbon"?

| Answer) Thanks for your reminder or thank you for your commenting. Indeed, this expression is not accurate. It has been changed into "ribbon".

Figure R1. Lateral size distribution of ribbons width ranging from 0 to 40 μm(a) and 5μm (b).

Furthermore, 85 ribbons were counted, and their width distribution is shown in Figure R1. From the statistics, we can see the lateral size of 70% ribbons is less than 10 μm and that of 5.4 % (27%*20% =5.4%) is less than 1 μm. Therefore, a small portion of ribbons can be called ‘nanoribbon’ while lateral size of most ribbons are more than 1 μm.

We have made corresponding corrections in the revised manuscript.

(line 26~28 page 7)

Q2: The authors mentioned that bulk antimony can be viewed as ABC stacking of monolayer antimonene. What is the stacking order of the few-layer antimonene in this manuscript? In addition, the crystal constants of few-layer antimonene should also be given.

Answer) Thanks for your comments.

1) As we know, stacking order significantly affects the properties of 2D materials. In our manuscript, we have added the description into the TEM characterization.

For comparison, with the same microscope parameters, the simulated image of few-layer antimonene with ABC stacking is shown in Figure S8 (Supplementary Information).

Obviously, there is very good agreement on the arrangement of bright points between the experimental and the simulated atomic images, which confirms the ABC stacking of the synthesized antimonene layers.

Besides, comparing the line scanning profiles of experimental image with those taken directly from the calculated (dash line) images along the dashed lines, we can also find atomic image obtained experimentally in good agreement with the simulated ABC stacking.

(Line 3~7 Page 10)

2) As to the crystal constants, β -Sb possesses a rhombohedral structure though theoretical workers usually view it as hexagonal structure. And relative parameters ($a=b=c=4.418$, $\alpha=\beta=\gamma=56.979^\circ$) is added into Figure 2a of the new manuscript.

$C_s = -0.013 \text{ mm}$, $\Delta f = 7 \text{ nm}$

Figure R2. Simulated atomic images and line scanning profiles of antimonene with different stacking under the condition of $C_s = -13 \mu\text{m}$ and $\Delta Z = +7 \text{ nm}$ ((above) and TEM characterization part shown in Figure 3 (down).

Q3: The antimonene sheets in Figure 1g and Figure S2 are the same sample, why the thickness of the former is ca. 4 nm, while the later is 1 nm? Did you put the picture wrong in Figure S2?

Answer) Thanks for your commentsreminding. We errantly put Figure 1g in Figure S2 carelessly and we have just deleted the ~~the~~ wrong picture.

Q4: The authors pointed out that the absence of dangling bonds on the substrate surface is critical for the successful growth of antimonene polygons. However, one antimonene sheet grown on mica in **Figure S3** shows irregular shape. Is it oxidized or formed by some other reasons?

Answer) We appreciate the reviewer's comments. Raman spectra, AFM files and TEM-EDS files are added in the latest manuscript, confirmed the reliability and stability of our experiments and instruments. Instead of oxidization, we attribute the irregular shape to growth kinetics. Normally, the growth of 2D crystal can be divided into two stages: nucleation and growth. As shown in **Figure 1h(the latest manuscript)**, samples, obtained after one-hour growth, usually possess a average lateral size of 6 μm . However, the sample possesses the irregular shape with a ca. 100 nm lateral size. It may be attributed to the poor crystallinity, further resulting in growth arrest.

Q5: The authors mentioned that the migration energy barrier of antimony atoms on mica substrate is very small, resulting in a high migration and fast growth rate. Are there any references or calculation results to support this piont? In addition, after 60-min growth, the lateral size of the sample is only ca. 10 μm . Such growth rate can hardly be called "fast".

Answer) We appreciate the reviewer's comments.

1) As to the "low migration barrier" and "fast growth rate", Yes, they have been considered as two features of Van der Waals Epitaxy. For example, in **reference 23**, the authors clearly

pointed out that the absence of dangling bonding resulted a low interaction between adatoms and substrate. Therefore, the migration barrier is 'low' and growth rate is 'fast'.

vdWE growth on mica sheets by PVD. Since mica provides a completely passivated surface that can interact with Te atoms through weak van der Waals forces. Consequently, high migration rate of Te adatoms on mica surface may lead to the faster growth of 2D Te nanoplates along the lateral direction.

Figure R2. Descriptions regarding “low migration barrier” and “fast growth rate” in reference 23.

2) As to the accurate growth rate of antimonenes, we implemented additional time-dependent growth experiments, as shown in **Figure S4**. According to **Figure S4**, the growth rate of antimonenes is about $0.5 \text{ um}\cdot\text{min}^{-1}$ and actually the growth has been finished in the first 10 minutes even for longer durations.

We have added the clear description on the growth rate in the revised manuscript .

(line 10~22 Page 8)

Q6: The characterization of the stability of the antimonene sheet is not convincing only by Raman analyses. Optical, AFM and TEM-EDX measurements of the freshly-made antimonene sheet and that stored one month should be given.

Answer) We appreciate the reviewer's comments.

Optical, AFM and TEM-EDX was employed to characterize the stability as the reviewer advised. Both the measurements indicates the stability, detailedly illustrated in 'High stability of multilayer antimonene'.

This suggestion is very helpful. These additional experiments were implemented as shown in **Figure 4**.

And, we have added the clear description on the stability in the revised manuscript in the section **'High stability of multilayer antimonene'**.

Q7: In the calculation part, the unit in the text is "ps", while it is "ns" in Figure 4. What is the real unit?

Answer) Thanks for your comments. The correct unit is 'fs' and we have corrected the description **(line 26 Page 13, line 28 Page 13).**

Q8: The V_{ds} in the inset of Figure 5c is 0.05V, while it is 0.5V in the text. Which one is the real measurement condition?

Answer) Thanks for your comments. The correct description is '0.05 V'.

(line 27 Page 14, line 6 Page 28)

Reviewer #2

1. Actually, this study is not the first experimental work of antimonene layers since Lei et al. (J. Appl. Phys. 119, 015302, 2016) have grown antimonene on Sb_2Te_3 and Bi_2Te_3 substrates in the beginning of this year.

Answer) We appreciate the reviewer's reminder. Strictly, the JAP paper is indeed the first experimental work of antimonene. And, we added the paper into the citations of our paper.

Lei et al. firstly achieved the synthesis of antimonene via MBE(molecular beam epitaxy). Samples obtained via MBE and research the band structure of monolayer antimonene. The spin-orbit coupling phenomenon was observed in monolayer antimonene. The work is worth a good evaluation.

However, plenty of necessary measurements, such as Raman spectroscopy, AFM and TEM, were absent, which are actually critical ~~was bad~~ for further study on antimonene. ~~and greatly prevented the work from higher evaluation.~~

In our manuscript, we reported the physical vapor transport synthesis of multilayer antimonene, together with a systematic characterizations.

Anyway, we believe that both of our research will promote the further research and application of antimonene.

2. The authors mentioned that van der Waals epitaxy can be achieved even when the lattice constant mismatch is as high as 50% and the epitaxial layer is completely relaxed without excessive strain. If the interaction between the epitaxial layer and the substrate is totally van der Waals force, authors can only say that it is without excessive strain. But they cannot say that the epitaxial layer is completely relaxed. The “relax” of epitaxial layer is defined as the emergence of threading dislocations from the interface through the epitaxial layer in order to release the excess strain energy. The excess strain energy is caused by the lattice mismatch between the epitaxial layer and the substrate as the critical thickness of epitaxial layer is proceeded. If so, the interaction between the epitaxial layer and the substrate should be chemical bonding. However, the authors claimed that the interaction is totally van der Waals force. Since the van der Waals epitaxy can overcome the limitation of lattice mismatch in conventional heterogeneous epitaxy, the scale of epitaxial thin film should be reached to wafer level. However, the scale of antimonene layers in this study is less than 10 μm . In fact, the quality and scale of heterogeneous epitaxial materials depend on several factors, not only lattice mismatch. Due to the micrometer

scale, it might not be the van der Waals epitaxy. If the authors believe that the interaction is totally van der Waals force, could they provide further evidence? For example, the XPS analysis could be used to confirm that there are not any chemical bonds between Sb and other elements in the substrate. Moreover, the cause of the limited scale of antimonene layers needs to be well explained, if the totally van der Waals force between the antimonene layers and mica substrate is proved.

Answer) We appreciate the reviewer's comments.

As the reviewer reminded, we did mix up some basic concepts. The “relax” in epitaxial layer doesn't corresponds to van der Waals epitaxy. According to the kind suggestion, we have corrected out initial expression to avoid the confusion. **(line22 Page 6)**

And, the interaction force between Sb epitaxy layer and mica is van der Waals force undoubtedly, reasons listed as follows:

Firstly, mica is completely passivated by F ions. Therefore, there is no possibility to form chemical bonding between epitaxial layer and mica.

Figure R4. Figure 3e in **reference 23**. Surface atomic structure of mica, passivated by F ions.

Secondly, lots of research on van der Waals epitaxy based on mica was published and typical papers have been cited in the manuscript, such as **Referemnce16, 18 and 23** in the manuscript.

- 16 Koma, A. Van der Waals epitaxy—a new epitaxial growth method for a highly lattice-mismatched system. *Thin Solid Films* **216**, 72-76 (1992).
- 18 Zhou, Y. *et al.* Epitaxy and photoresponse of two-dimensional GaSe crystals on flexible transparent mica sheets. *ACS nano* **8**, 1485-1490 (2014).
- 23 Wang, Q. *et al.* Van der Waals Epitaxy and Photoresponse of Hexagonal Tellurium Nanoplates

on Flexible Mica Sheets. *ACS nano* **8**, 7497-7505 (2014).

Reference 16 firstly claimed the van der Waals epitaxy on the mica. The growth of several layered materials was mentioned for demonstration purposes. Later, a series of research based on epitaxy on mica was conducted. And, van der Waals epitaxy on mica is explicitly pointed out in **Reference 18** and **23**.

Furthermore, as suggested, to confirm the van der Waals epitaxy, XPS was employed. As shown in **Figure S2a**, peaks of Sb 3d_{3/2} and Sb 3d_{5/2} appears at ca.528 eV and 538 respectively, in good agreement with nonvalent Sb while O 1s peak at ca.532 eV is attributed to mica. It's proved that there are not any chemical bonds between Sb and other elements on the substrate.

Another intuitionistic evidence is the morphology difference between samples fabricated on different substrates, which we have mentioned in the manuscript. As shown in Figure R5, antimony samples grown both SiO₂/Si and sapphire possess irregular shape and higher thickness while antimonene layers have well-defined shapes. These differences is attributed to surface induction.

Figure R5. Antimony grown on different substrates.

We also noticed that there is a size limitation during the van der Waals epitaxy growth of 2D crystals on mica and epitaxial layer can't fully cover the substrate. It has never been pointed out by clairvoyants. Herein, we'd like to share our speculation drawn from paper reading and analysis.

As the reviewer pointed out, crystals growth on mica, usually show a limited lateral size, which is a common phenomenon in this field, although mica was applied widely to van der Waals epitaxy. This confused us too.

Figure R6. GaSe epitaxy on mica without treated (Left) and with O plasma treated (Right) in Reference 18.

In reference 18, Yubing zhou et al. showed position control growth of GaSe on mica treated by oxygen plasma. Interestingly, on the surface of treated position, GaSe crystals covered fully. Meanwhile, crystals, grown on activated surface, seems to be much thicker than those grown passivated substrates. Noticing this, it's speculated that surface activation may result in a sharply increased adsorption of vaporized precursors while a totally passivated substrate strictly limits the adsorption of atoms, further resulting in size-limitation and poor coverage.

3. The authors mentioned that the space between two sets of lattice fringes is measured to be 0.204 nm, corresponding to the (100) plane of antimony. The angle between two fringes is measured to be 120° , characteristic of hexagonal structure. However, grey antimony possesses rhombohedral structure. Its d spacing of (100) plane group should be 0.354 nm, the included angle between (100) and (-1-10) planes of it should be around 54° (or 126°) in theory. The material identification must be precisely checked.

Answer) We appreciate the reviewer's comments.

Figure R7. Base vectors selection in hexagonal structure and rhombohedral structure(left) and Base vectors in **Reference 34**.

Firstly, as shown in Figure R7, hexagonal structure and rhombohedral structure can interconvert via different selection of base vectors in crystal.

As the reviewer pointed out, grey antimony possesses rhombohedral structure. However, up to now, experimental research is rare and a great portion of research on antimony is theoretical calculation. Usually, theoretical workers can conveniently build primitive cell when grey antimony is viewed as ABC stacking of monolayer antimonene along $[111]$ axis with \mathbf{a} , \mathbf{b} and \mathbf{c} as the base vectors, shown in Figure R5. And, **Reference 34** is a typical example. Initially, we adopted this opinion to calculate the crystal parameters. Obviously, such an operation results strict confusion since grey antimony is known to possess rhombohedral structure. This has been corrected in rewritten '**Atomic scale microstructure of antimonene single-crystal polygons**' section. (line9 Page 9~line 11 Page 10)

Also, the reviewer pointed out some incorrect calculations in initial manuscript, which have been corrected as suggested, too.

For the absence of crystal parameters of rhombohedral structure in powder diffraction files(PDF) database, we calculate the diffraction spot of grey antimony along $[111]$ axis. According to the simulated diffraction spots, we remark the SAED in the manuscript and rewrite TEM

characterization section, as suggested. And, both of (100) and (-1-10) can not be found along [111] axis.

Figure R6. Simulated diffraction spot of grey antimony(along [111] zone axis).

4. The crystal structure of β -Sb belongs to rhombohedral lattice and that of α -Sb belongs to orthorhombic lattice. The description, “the image clearly demonstrates a typical rhombic structure (along [001] axis), completely different from the hexagonal structure of β -phase antimonene,” may confuse the readers. The descriptions regarding the crystal structures in the manuscript need to be rewritten for clear expression.

Answer) We appreciate the reviewer’s comments.

In the initial manuscript, the we selected **a**, **b** and **c** as base vectors and viewed grey antimonene as a hexagonal structure, a confusing description regarding the crystal structure.

As the reviewer suggested, the descriptions regarding the crystal structure have been rewritten for the rhombohedral structure of grey antimony. (line9 Page 9 ~line 11 Page 10)

5. In thermodynamic simulation, the authors cannot only consider the difference of exothermic energy (Q), which is equal to enthalpy (H), between grey antimony and black phosphorus oxidations. The entropy should be also considered. Therefore, the difference of Gibbs free energy (G) between grey antimony and black phosphorus oxidations should be estimated in order to make sure which oxidation occurs preferentially.

Answer) We appreciate the reviewer's comments. According to the suggestion, in the thermodynamic simulation we have taken entropy into account. The calculated difference of Gibbs free energies (ΔG) at room temperature of 298.15 K are:

Both have positive value of ΔG , 59.394 and 15.993 kcal/mol, respectively. Though the enthalpy of the two reactions is less than zero, showing an exothermic process, the entropy of those reactions is reduced. From the calculated results, it suggests that both reactions are not spontaneous reactions at room temperature, and the product of $\text{O}=\text{Sb}_{2\text{D}}=\text{O}$ is more difficult to generate than that for $\text{O}=\text{P}_{2\text{D}}=\text{O}$ because of the higher value of ΔG . Thus, the calculated results and our strong experimental evidence also confirmed that the oxidation of 2D grey antimonene cannot occur preferentially, which is more difficult than that of black phosphorene.

Figure R7. Thermodynamic simulation of Gibbs free energy (G) between greyantimony and black phosphorus oxidations

6. Essentially, antimony is a semimetal unless its thickness reduces under bilayer in theory. The antimony films obtained in this study cannot be called “antimonene” which is single layer antimony. It should be called multilayer antimonene or antimonene layers. It is unnecessary to use a known semimetal material as the channel of transistor to prove that it is a semimetal. The scale of the antimonene layers obtained in this study is too small to be commercialized for industrial device fabrication.

Answer) We appreciate the reviewer’s comments. Samples in our research are indeed more than bilayer. And, ‘multilayer antimonene’ is more accurate. We have corrected it in the latest version manuscript.

The reviewer mentioned that antimony is a semimetal unless its thickness reduces under bilayer in theory and it is unnecessary to use a known semimetal material as the channel of transistor to prove that it is a semimetal, consistent with the calculation in our previous paper(**Reference 10**).

Firstly, one purpose of this test to confirm that it’s truly β -Sb instead of α -Sb since α -Sb, as a isologue of black phosphorous, is expected to be stable and semiconducting. Therefore, the test can further confirm the phase of our sample.

Furthermore, in fact, Hsu-Sheng Tsai et al. claimed that β -As was a semiconductor with a 2.3 eV bandgap early this year(Tsai H S, Wang S W, Hsiao C H, et al. Direct Synthesis and Practical Bandgap Estimation of Multilayer Arsenene Nanoribbons[J]. *Chemistry of Materials*, 2016, 28(2): 425-429.). There is a urgent necessity to verify the electronic properties of β -Sb since it’s a isologue of β -As. Actually, during our revision, they claimed β -Sb was similar to β -As and has a bandgap in another paper(Reference 36).

36 Tsai, H. S., Chen, C. W., Hsiao, C. H., Ouyang, H., & Liang, J. H. The Advent of Multilayer Antimonene Nanoribbons with Room Temperature Orange Light Emission. *Chemical Communications* (2016).

So, electrical tests in our paper are quite important to clarify the potential confusion.

As to the present sample scale, just as the reviewer pointed out, it is indeed far from industrial application. The demonstration shown in our paper was just based on the material properties to show a potential application. And, looking back on the the growth of 2D materials, we can find that the successful synthesis of 2D materials, responsible for practical commercial application, wasn't usually realized in a short term.

As the reviewer expected, we will devote more efforts to grow samples with higher quality and larger scale for further research and application.

7. The authors mentioned that the antimonene layers have potential to be used for the applications of the transparent conductive electrode. However, the scale of antimonene layers obtained in this study is too small to be a thin film. In the bending test, how do the authors confirm that the bending effect is indeed exerted on the antimonene layers with the scale less than 10 μm , since the scale of mica substrate is centimeter. In addition, the authors must enlarge the scale of antimonene layers up to centimeter scale for real applications.

Answer) We appreciate the reviewer's comments.

1) As the reviewer mentioned, antimony polygons was too small for us to confirm if the bending effect exerted on the antimony polygons. Therefore, we make some adjustments.

Mica substrate was attached onto the surface of iron rod to confirm that the bending effect is equably exerted on the whole substrate. Subsequently, we retest the conductivity change as a function of bending times and drew a more strict conclusion. As shown in **Figure 5d**, bending test affects little on the conductivity.

2) Just as the reviewer pointed out, the present sample scale can't satisfy real application. In fact, we urgently want to enlarge the sample scale, as the reviewer expected.

8. The authors need to correct many typos in the manuscript and the writing should be polished by a native English speaker.

Answer) We appreciate the reviewer's comments. We checked the manuscript carefully and correct the typos, as the review pointed out.

“revealled”-----”revealed” **(line 27, page 12)**

“especiall”-----”especially” **(line 30, page 12)**

“sysstem”-----”system” (line 6, page 13)

“stuctures”-----”structures” (line 8, page 13)

“eqation”-----”equation” (line 9, page 13)

“respectivley”-----”respectively” (line 11 page 13)

“halfnium”-----”hafnium” (line 6, page 14)

“ahalogen”-----”halogen” (line 13, page 17)

Reviewer #3

i) Page 7. The authors state that "The migration energy barrier of antimony atoms on mica substrate is very small, which results in a high migration rate along the mica substrate and a fast lateral growth of 2D antimonene polygons."

Is there any supporting references or data for this statement?

Answer) We appreciate the reviewer’s comments. In fact, mica is widely applied as the van der Waals epitaxy substrate. Compared with other substrates, smaller atom migration energy barrier on mica is clarified in many papers, some of them listed in the manuscript:

- 16 Koma, A. Van der Waals epitaxy—a new epitaxial growth method for a highly lattice-mismatched system. *Thin Solid Films* **216**, 72-76 (1992).
- 17 Ji, Q. *et al.* Epitaxial Monolayer MoS₂ on Mica with Novel Photoluminescence. *Nano Letters* **13**, 3870-3877 (2013).
- 18 Zhou, Y. *et al.* Epitaxy and photoresponse of two-dimensional GaSe crystals on flexible transparent mica sheets. *ACS nano* **8**, 1485-1490 (2014).
- 23 Wang, Q. *et al.* Van der Waals Epitaxy and Photoresponse of Hexagonal Tellurium Nanoplates on Flexible Mica Sheets. *ACS nano* **8**, 7497-7505 (2014).

The successful growth of SnSe₂, GaSe and Tellurium nanoplates are all good examples of van der Waals epitaxy. In a **reference 16**, Atsushi Koma raised up that the mica surface with no dangling bonding promoted the van der Waals growth of layered materials. As typical papers in van der Waals epitaxy growth of 2D materials, **reference 17, 18 and 23** all pointed out that low barrier energy results in a high migration rate, because the interaction between atoms and mica is van der Waals force, instead of chemical bonding.

Furthermore, as suggested, to confirm the van der Waals epitaxy, XPS was employed. As shown in **Figure S2a**, peaks of Sb 3d_{3/2} and Sb 3d_{5/2} appear at ca. 528 eV and 538

respectively, well fitting with nonvalent Sb while O 1s peak at ca.532 eV is attributed to mica. It's proved that there are not any chemical bonds between Sb and other elements in the substrate. **(line 15~21 Page 7)**

ii) Figure 4. Did the authors run ab-initio calculations up to 2000 ps? In the main text, the authors state that the simulation is up to 2000 ps, while in Figure 4, it is 2000 fs.

Answer) We appreciate the reviewer's comments. The ~~right~~ unit is actually 'fs'. In our new manuscript, we have corrected the ~~mistakebug~~. **(line 26 Page 13; line28 Page 13)**

iii) Reference 25 predicts α and β phases to be degenerate. Can the authors comment on it?

Answer) We appreciate the reviewer's comments. In this question, there are missing "two word": *iii) Reference 25 predicts ? and ? phases to be degenerate*. In our opinions, we guess that this question should be like: "iii) Reference 25 predicts γ and δ phases to be degenerate. Can the authors comment on it?". In Reference 25, Prof. Pandey group studied antimonene, with four interesting phases, α , β , γ and δ phases. In group V, α and β are typical phases. In fact, the counterpart bulk material of α and β group V monolayer, such as phosphorene, arsenene and antimonene, are very stable phases among their own allotropes. For example, the counterpart bulk materials of β -arsenene, β -antimonene, and β -bismuthene are rhombohedral layered gray arsenic, gray antimony, and β -type bismuth under normal conditions with the same space group. To our knowledge, till now two types of layered phases experimentally exist in the group V phases, namely, the bulk α -P, α -As, β -As, β -Sb, β -Bi. So, for monolayer it seems that α and β are more stable than other phases (γ and δ phases). Of course, our previous work and Pandey's work together confirmed this point. (Angew.Chem, 2015, 54, 3112; and Angew.Chem, 2016, 128, 1698. ACS Appl. Mater. Interfaces, 2015, 7, 11490.)

iv) The results of a recent paper for oxygen in phosphorene should be bench-marked with the results obtained previously (Wang et al. 2D Mater. 3, 025011, 2016).

In the present study, the calculated dissociation energy of O2 on phosphorene is high. Did the authors use triplet or singlet O2 to get the dissociation energy?

Answer) We appreciate the reviewer's comments. Our results for oxygen in phosphorene have been bench-marked with the results obtained previously (Wang et al. 2D mater. 3, 025011, 2016). The value reference has been cited in the revised manuscript. All references are corrected in the text in sequential order throughout the whole manuscript.

In our old manuscript, we use singlet O₂ to get the dissociation energy, and the calculated dissociation energy of O₂ on phosphorene is high. According to your kinder suggestion, we take into account the triplet O₂ to further study the dissociation energy. The dissociation energy of O₂ on phosphorene is -4.15 eV/O₂ (In the old manuscript, -4.80 eV/O₂). We also recalculated the dissociation energy of O₂ on antimoene, which is -1.01 eV/O₂ (In the old manuscript, -1.62 eV/O₂). Thus, Our results are in good agreement with the previous reports (Wang et al. 2D mater. 3, 025011, 2016). Relevant results and references have been added in our new manuscript. [Note that the triplet state of O₂ are used to calculate the dissociation energy, and the method has been well confirmed in phosphorene oxidation (Wang et al. 2D mater. 3, 025011, 2016).]

v) The dissociation energy of O₂ on antimonene is calculated to be -1.62 eV/molecule, which implies that O₂ prefers to dissociate on antimonene. XPS measurement could confirm whether O₂ is dissociated or not. Can the authors address this discrepancy?

Answer) We appreciate the reviewer's comments.

Calculations indicates the tendency that O₂ prefers to dissociate on antimonene,

To verify the stability of antimonene experimentally, XPS was employed to measure sample stored for one week after sample synthesis. In **Figure S2a**, peaks at 528.4 eV and 538 eV correspond to Sb 3d₅ and Sb 3d₃, respectively, which is attributed to nonvalent antimony and consistent with reference 34. Its stability was further confirmed. Also, TEM-mapping in **Figure 2** indicates the similar conclusion.

We think that the oxidation may happen according to the calculation, but the reaction is quite slow. Storage in atmosphere isn't harsh enough to result in oxidization, consistent with XPS spectrum, while TEM-EDX indicates sight oxidization, properly caused by heating operation(baked at 60 °C, 80 °C and 150 °C) during transferring nanosheets onto copper grids.

iii) Pl correct the citation for references and figures in the main text.

(a) Reference 13 -> Reference 25. Reference 25 -> Reference 13,25.

(b) Page 10. "Raman spectra of typical antimonene polygons stored in air for one month after sample synthesis are shown in Figure S5."

(c) Page 8. "The selected area electron diffraction (SAED) pattern shown in the inset of Figure 3c further confirms the hexagonal structure and its single crystal feature..."
Figure 3c -> Figure 2c.

Answer) We appreciate the reviewer's comments.

(a) There are indeed some citation errors, which have been corrected in the latest version according to the suggestion. **(line 29 page 4, line 26 page 6)**

(b) We did make a mistake. This error was corrected. According to the kind comments, We have change 'Figure S5' to 'Figure 4a'.

(c) We did make a mistake. This error was corrected. According to the kind comments, We have change 'Figure 3c to 'Figure 2e' inset.

(iv) Text: DFT calculations are not 'ab-initio' calculations. ab initio -> first principles

Answer) We appreciate the reviewer's comments.. In the new manuscript, we have changed "ab-initio" into "first-principles".

Reviewers' comments:

Reviewer #1 (Remarks to the Author):

The authors have addressed my comments and suggestions properly, therefore, I would like to recommend the publication of this manuscript in Nature Communications.

Small questions:.

- 1) The authors should give solid characterization results to prove that the 1-nm sample shown in Figure 1h is indeed antimonene. In addition, no scale bar was given in Figure 1h.
- 2) In Figure 2a, no unit was given for the crystal parameters: $a = b = c = 4.418$.
- 3) Please check the labels of XPS Sb 3d peaks shown in Figure S2.
- 4) The representation of zone axis in Figure S6 should be changed from (001) to [001] and from (111) to [111].

Reviewer #2 (Remarks to the Author): report attached.

Reviewer #3 (Remarks to the Author):

The authors have satisfactorily addresses the comments.

* Although we cannot offer to publish your manuscript in Nature Communications, the work may be appropriate for another journal published by Springer Nature. If you wish to explore suitable journals and transfer your manuscript to a journal of your choice without having to re-supply manuscript metadata and files, please click on

<http://mts-ncomms.nature.com/cgi-bin/main.plex?el=A3S3ize6B6DrYq1X3A9ftdTZSyBC2pF7fon1x2EgOzwZ>

For more information, please see our http://www.nature.com/authors/author_resources/transfer_manuscripts.html?WT.mc_id=EMI_NPG_1511_AUTHORTRANSF&WT.ec_id=AUTHOR >Manuscript Transfer FAQ page.

Part 1. Summary of Comments from Reviewers

Reviewer #1: (Recommendation of Acceptance)

The authors have addressed my comments and suggestions properly, therefore, I would like to recommend the publication of this manuscript in Nature Communications. Some small questions need further answer.

Reviewer #2: (Reject of Publication)

Reviewer #3: (Recommendation of Publication)

The authors have satisfactorily addresses the comments.

Part 2. Response to Comments from Reviewers

2.1 Response to Reviewer #1

Comment 1. The authors should give solid characterization results to prove that the 1-nm sample shown in Figure 1h is indeed antimonene. In addition, no scale bar was given in Figure 1h.

Answer. We appreciate the comment. The scale bar '50 nm' has been added in the revised manuscript. Actually, it's hard to obtain the TEM-EDS and AFM data of a certain nanosheet at the same time. However, tiny sheets abound, which offers great convenience to measure the chemical composition. And, additional TEM-EDS experiments (Figure S4) confirm the tiny sheets to be antimony.

Comment 2. In Figure 2a, no unit was given for the crystal parameters: $a = b = c = 4.418$.

Answer. We appreciate the comment. The unit ' \AA ' has been added to Figure 2a in the revised manuscript.

Comment 3. Please check the labels of XPS Sb 3d peaks shown in Figure S2.

Answer. We appreciate the comment. The labels have been checked and corrected as below.

Comment 4. The representation of zone axis in Figure S6 should be changed from (001) to [001] and from (111) to [111].

Answer. We appreciate the comment. The representation of zone axis in Figure S6 has been revised in the revised manuscript.

2.1 Response to Reviewer #2

Coment 1. It is true that this work is not the first work about the synthesis of

antimonene!! I think it should not be published in Nature Communications, since it only accepts the first pioneering result. Also, the results in this work are very poor. The results obtained by high-level instruments such as MBE, STM and ARPES are much accurate enough for publishing in Nature series journals (e.g. *Nat. Mater.* **9**, 315-319 (2010)). Even though lots of studies only utilize STM for analysis, they still can publish the only STM results in high impact factor journals because of the accuracy of STM.

Answer. We report the PVT synthesis of antimonene while The JAP paper reported the synthesis of antimonene via MBE.

First significant breakthrough is our method. As we know, PVT, together with the generally similar CVD, wins much popularization and broad recognition for combining crystal quality, large scale production and low cost while expensive and complex instruments usually results in hindered advance of both scientific research and real application. A typical example is stanene fabricated via MBE (*Nature Materials* 14, 1020–1025(2015)). Few experimental research follows the famous NM paper, illustrating how a complex and expensive method hindered the advance of stanene research. Anyway, a report of highly-popularized method will surely raise the great attention and promote the both scientific research and real application.

Then, samples, large enough for normal device fabrication, are achieved, which dramatically breaks through the JAP paper. By contrast, the JAP paper only fabricated domain with a few hundred nm scale, which is far from most research and incurs comparatively low evaluation.

Moreover, abundant original data are listed detailedly in the following comparison table.

Detailed Comparison differences are illustrated in the table below.

	The JAP paper	Our Manuscript
Technology	MBE	PVT or PVD

Lateral Size of Single Domain	Not mentioned the the paper; Usually several hundred nanometers (Nature Materials 14, 1020–1025(2015)) Limited to same specific scientific research, such as ARPES in the paper.	Nanosheets: Several micrometers Ribbons: Width: hundreds nanometer~More than ten micrometers Length: Tens micrometers. Satisfied to most research, just like Graphene and MoS ₂ .
Thickness& Evidence	Monolayer, no other experimental evidence except for ARPES data.	Multilayer confirmed via AFM, Raman spectroscopy and HRTEM
Research Contents	Materials fabrication via MBE; Phase identification via LEED; Electronic structure via ARPES.	Crystal growth via PVT; Growth mechanism research via XPS and AFM; Phase identification via TEM, EDS, Raman Spectroscopy; Thickness-Dependent Raman Spectra; Air stability research via EDS, XPS, AFM; Thickness-dependent Transmittance Electrical conductivity test; Application illustration.
Difficulty of technology	High Difficulty	Low Difficulty
Cost	High	Extremely Low
Popularity	Low	Much Higher

Table 1. Novelty Explanation via comparing the JAP paper with ours.

Comment 2.1. How do authors totally make sure that mica is completely passivated by F ions?

Answer. We appreciate the comment. This information is mentioned in Reference 23 and the crystal structure of mica is illustrated in Figure 3c in Reference 23.

Comment 2.2. The Ref. 16 only provides several unclear RHEED patterns. I do not agree their claim and the following related articles are incredible. They cannot increase the lateral dimension of the thin films. There are other factors limiting the lateral dimension during growth. Please note that the literatures are not always

correct!!

Answer. We appreciate the comments. However, these typical papers are well recognized in vdWe growth of crystals and corroborate each other, which offer a new approach to fabricate 2D crystals. And, we think that it's improper to deny these solid achievements without abundant and solid experimental evidences.

Of course, as the reviewer mentioned, such a method usually results in a limited scale, which is worth further study.

Comment 2.2. If there are not any chemical bonds between Sb film and other elements in the substrate, why can the lateral dimension of the thin film not be increased? Authors did not answer my question.

Answer. We appreciate the comments. In **reference 18**, Yubing zhou et al. showed position control growth of GaSe on mica treated by oxygen plasma. Interestingly, on the surface of treated position, GaSe crystals covered fully. Meanwhile, crystals, grown on activated surface, seems to be much thicker than those grown passivated substrates. Noticing this, it's speculated that surface activation may result in a sharply increased adsorption of vaporized precursors while a totally passivated substrate strictly limits the adsorption of atoms, further resulting in size-limitation and poor coverage. According to this, we attributed the limited scale to low adsorption rate resulting from the chemically passivated surface.

Comment 2.3. No matter what kind of substrate used in this work, the qualities of Sb are very poor for real application.

Answer. We appreciate the comments. In our manuscript, sample scales of monocrystalline polygons and ribbons are up to ca.8 micrometers and 50 micrometers, respectively, which can be treated as a great breakthrough in 2D material.

Of course, sample scale satisfying the needs of real applications is important during our research. Actually, even now, few 2D materials can strictly meet the need of real application, which is common during our research. And, from my point of view, real application isn't the only purpose of scientific research and the sole criterion to evaluate others' research.

Comment 2.4. Even though the mica is not treated by oxygen plasma, the lateral

dimension is still limited. This quality is also very poor for real applications.

Answer. We appreciate the comments. If the Reviewer #2 really read Reference 18, he/she will find our explanation is reasonable.

And, we insist that real application isn't the only purpose of scientific research and the sole criterion to evaluate others' research.

Comment 3. The crystal structure of “gray” arsenic is as same as that of “gray” antimony. The only difference is the lattice constant, implying that the atomic positions of them are the same. Authors should read the reference as below.

J. Appl. Cryst. (1969). 2, 30

Answer. We appreciate the comments. Lattice parameters of β -As indeed mentioned in the paper, which should be a good reference, if we indeed gave a wrong labeling.

However, **the selection of base vectors and labeling details in the recommended paper is almost the same with our initial manuscript.** Such a query is unreasonable from the beginning.

Besides, our labeling has been carefully checked and has a good consistence with Reference 31 and 32.

Comment 4. It may be okay.

Answer. We appreciate the recognition from the reviewer for our hard working and detailed explanation.

Comment 5. Actually, black phosphorus is quite unstable under atmosphere. Its ΔG of oxidation may be negative. The simulation results should be checked again.

Answer. We appreciate the comments. The simulation results have been well checked. In a degree, it explains our opinion.

Comment 5. 1. The case of Tsai et al. is totally different from that of authors. Authors have confirmed the crystal structure of their Sb flakes. It belongs to rhombohedral structure (β -Sb) not orthorhombic one (α -Sb). The FET fabrication is totally unnecessary.

Answer. We appreciate this comments. The reviewer mentioned the case is different from ours. However, current published papers haven't given a experimental explanation of the PL spectra of As and Sb film in Tsai' papers. So, we think it's proper to show these data although such results are not breathtaking enough.

Comment 5. 1. Lei et al. and Tsai et al. have already experimentally shown the potential of antimonene for applications. The antimonene layers obtained by their groups are with better quality and larger scale than those obtained by authors. So far, the scale of antimonene layers achieved by authors is too small to be accepted for publication.

Answer. We appreciate the comments.

Firstly, we repeat our insistance that that real application isn't the only purpose of scientific research and the sole criterion to evaluate others' research.

Then, we notice that '**the potential of antimonene for applications**' of Tsai' work wins the recognition from Reviewer #2. However, our application illustration is faced with strict requirements of '**real applications**' from the reviewer, which confuses us.

Comment 6.

1) It may be okay.

2) I **think** that every group **can** achieve this scale; it does not have any breakthrough.

Answer.

1) We appreciate the recognition from reviewer for our adjustment.

2) We clearly disagree on such a comment. As far as we know, these results are only realized in our group up to now, which is not consistent with the expectation of reviewer 2. **Can ≠ do**. In fact, the threshold of mechanic exfoliation of graphene is so low that everyone can grasp the skills. However, nobody fabricated graphene until Andre Geim and Konstantin Novoselov did.

Comment 8. It is okay.

Answer. We appreciate the comment.

Comment 9. In a word, this work does not have any novelties and is not practical. **I do not think it can be published in any journals.**

Answer. We appreciate the comment. But, we can't agree on it at all.

1) Novelties of this manuscript have been detailedly explained again in the Table 1 in the beginning part.

2) Here, we repeat that real application isn't the only purpose of scientific research and the sole criterion to evaluate others' research.

REVIEWERS' COMMENTS:

Reviewer #1 (Remarks to the Author):

After carefully checking the point-by-point response and the revised manuscript, I think that the authors have addressed most of the issues raised by Referee #2. Although the lateral size and quality of the antimonene layers are indeed not large/good enough for real applications as Referee #2 mentioned, the PVD or PVT method reported in this manuscript really provide a new possibility to grow such kind of materials, which I think is the main novelty of this work. In the response, moreover, the authors also emphasized the several advantages of PVD or PVT method over the existing MBE method, for example, the larger later size, easy to operate, and low cost. For the transparent conductive electrode applications mentioned in the manuscript, the PVD or PVT method should be more likely to achieve the goal than the MBE method. Based on the above, I think that this manuscript could be considered for publications in Nature Communications after the following comments are properly addressed.

1. In Figure S4 & S5, scale bars are missing.
2. In Figure S8, the zone axis of β -phase is not along [111] rather [001].
3. There are some inconsistencies between the reference number and the content cited in the manuscript. For example, "Despite the rapid progress of theoretical works about antimonene mentioned above, experimental studies on antimonene is still scarce^{31,34,36}," where the references should be "31,32,34".
4. In Figure 4b, why not use the identical antimonene flake to make the comparison for the as-prepared and the 30-day aged samples?
5. The size of the 30-nm-thick antimonene is ca. 5 μm , while the bending radius is ca. 5 mm, which is significantly larger than the sample. I doubt that the bending effect can hardly be applied to the sample.

Response to Reviewer #1

We appreciate your general recognition and positive evaluation. And, we revised the manuscript (**marked with a pink background**) according to the previous suggestions and reply to the queries as below.

1. In Figure S4 & S5, scale bars are missing.

Answer. Thank you for your careful reading. The scale bar '100 nm' has been added to the description of Figure S4.

The scale bars of the nucleation and growth images in Figure S5 are 500 nm and 5 μm , respectively.

2. In Figure S8, the zone axis of β -phase is not along [111] rather [001].

Answer. We appreciate the reviewer's comment. This question is involved with the rhombohedral structure of antimony.

Figure R1. (a) Base vectors of hexagonal structure and rhombohedral structure . (b) Base vectors in in-plane atom layers in Reference 32.

Usually, hexagonal structure (\vec{a} , \vec{b} , \vec{c} as base vectors) and rhombohedral structure (\vec{a}' , \vec{b}' and \vec{c}' as base vectors) can convert into each other via different selection of base vectors in crystal, as shown on **Figure R1(a)**. Though antimony possesses a rhombohedral structure, for convenience, researchers (e.g. **Reference 32**) usually regard antimony as a hexagonal lattice due to the similar layer structure with graphene, as shown in **Figure R1(b)**.

The labeling in our former manuscript was based on the rhombohedral structure (\vec{a}' ,

b' and c'), which is uncommon and may result in potential queries.

In the latest manuscript, we have revised the descriptions based on *the hexagonal structure*, according to the suggestion.

Revised portions are listed as below.

Line 19 Page 8~Line 27 Page 8;

the labeling and description of Figure 2a, 2e, S6b, S7 and S8.

3. There are some inconsistencies between the reference number and the content cited in the manuscript. For example, “Despite the rapid progress of theoretical works about antimonene mentioned above, experimental studies on antimonene is still scarce^{31,34,36},” where the references should be “^{31,32,34}”.

Answer. Thank you for your careful reading and pointing our mistakes. According to the kind suggestion. We have checked the manuscript and corrected the mistakes as below.

~~31,34,36~~-----^{31,32,34} (Line 3, Page 4)

~~34~~-----³¹ (Line 4, Page 4)

~~36~~-----³² (Line 6, Page 4)

~~36~~-----³⁴ (Line 10, Page 4)

4. In Figure 4b, why not use the identical antimonene flake to make the comparison for the as-prepared and the 30-day aged samples?

Answer. We appreciate your valuable comment. The suggestion did offer a more strict experimental scheme.

In fact, it's easy to find a Sb nanosheet and characterize via TEM and EDS while, after a 30-day aging in air, it's hard to find the identical flake. Therefore, we find another flake and draw a qualitative conclusion.

During our experiments, the flakes were transferred from a mica to the copper grid at the same time and the flakes with similar initial compositions all underwent a 30-day aging and were regarded to possess similar compositions.

Then, it's noticed that the oxygen content of the flake observed after a 30-day aging was still quite low, which indicates the high stability of antimonene layers, despite of no comparison.

Besides, other experimental evidences and simulated results, shown in (Figure 4a, 4c and 4d), except for EDS spectra, are offered to illustrate the stability of Sb layers.

Figure 4. Stability verification via experiments and simulations of antimonene compared with BP. (a) Optical images, AFM images and Raman spectra of antimonene layers before and after a 30-day aging. The scale bar is 2 μm . (b) EDS analysis of antimonene sheets randomly selected on the copper grid before and after a 30-day aging. (c) Atomic structure of phosphorene oxide ($\text{O}=\text{P}_{2\text{D}}=\text{O}$) and time-dependent snapshots of the configurations revealing interaction of O_2 with phosphorene. (d) Atomic structure of phosphorene oxide ($\text{O}=\text{Sb}_{2\text{D}}=\text{O}$) and time-dependent snapshots of the configurations revealing interaction of O_2 with antimonene.

5. The size of the 30-nm-thick antimonene is ca. 5 μm , while the bending radius is ca. 5 mm, which is significantly larger than the sample. I doubt that the bending effect can hardly be applied to the sample.

Answer. We appreciate the the reviewer's comment. The bending radius is truly significantly too large to for us to confirm the bending effect, if we only take the size difference into consideration.

Firstly, during the test, mica was attached onto the surface of iron rod to ensure the bend effect was equally applied on mica, as shown in Figure 5d. Since the

macroscopic bending exists, the bending effect is regarded to be applied on mica due to the smooth surface.

Figure R2. Results and experimental details of bending test (Figure 5d).

Then, the electric conductivity decrease (see pink arrow) after bending test was observed, indicating that the bending effect was truly applied to the sample and caused the change of electric conductivity. This phenomenon we observed is highly consistent with **Reference 23**, where authors also carried out a bending test. The Te flake, in Reference 23, possess a similar scale with the Sb sheet, which further confirms the feasibility of our test.

23 Wang, Q. *et al.* Van der Waals Epitaxy and Photoresponse of Hexagonal Tellurium Nanoplates on Flexible Mica Sheets. *ACS nano* **8**, 7497-7505 (2014).

Figure R3. The snapshot of Figure 5 in Reference 23. Bending tests result in a decreased electric conductivity, consistent with our results. The scale bar of Figure 5a (inset) is 1μm.

To address the potential doubts, we add the following description in the revised manuscript.

The antimonene/mica sample was attached onto the smooth surface of a iron rod

to ensure the bending effect was applied.’ (Line 27 Page 13 ~ Line 28 Page 13)